# Extraction and Characterization of Self-Assembled Collagen Isolated from Grass Carp and Crucian Carp

**DOI:** 10.3390/foods8090396

**Published:** 2019-09-06

**Authors:** Li He, Wenting Lan, Yue Wang, Saeed Ahmed, Yaowen Liu

**Affiliations:** 1College of Food Science, Sichuan Agricultural University, Yaan 625014, China (L.H.) (W.L.) (Y.W.) (S.A.); 2School of Materials Science and Engineering, Southwest Jiaotong University, Chengdu 610031, China; 3California NanoSystems Institute, University of California, Los Angeles, CA 90095, USA

**Keywords:** collagen, self-assembly, grass carp, crucian carp

## Abstract

Collagens were extracted from grass carp skin (GCC), grass carp scales (GSC), and crucian carp skin (CCC) using an acid-enzyme combination method, and their characteristics and self-assembly properties were analyzed. Electrophoretic patterns characterized all three as type I collagens. An ultraviolet analysis identified the optimal wavelengths for collagen detection, while a Fourier transform infrared spectroscopy analysis confirmed the triple-helical structure of the collagens. The GCC, GSC, and CCC had denaturation temperatures of 39.75, 34.49, and 39.05 °C, respectively. All three were shown to self-assemble into fibrils at 30 °C in the presence of NaCl, but the fibril formation rate of CCC (40%) was slightly higher than those of GCC (28%) and GSC (27%). The GSC were shown to form a more strongly intertwined fibril network with a characteristic D-periodicity. The fish collagens extracted in this study have potential applications in the development of functionalized materials.

## 1. Introduction

Collagen is the main protein in animal connective tissue, and it has been widely used as a food [1] and as an industrial functional material [2]. At least 28 types of collagen have been identified [3], each with a different molecular structure, amino acid sequence, and functionality. Type I collagen is the major structural component in human and animal skin, and it is the most extensively-studied collagen type [4]. Traditionally, type I collagen has been isolated from the skin and bones of terrestrial animals, such as swine, cattle, and poultry. However, religious prohibitions and concerns about the spread of bovine sponge encephalopathy, foot-and-mouth disease, transmissible spongiform encephalopathy, and avian influenza to humans have limited the use of collagens from these sources [5]. The skin and scales from many species of fish received attention [6]. In China, large quantities of fish processing material are disposed of as waste, accounting for as much as 50–70% of the original raw material [7]. The isolation of collagen from these wastes could provide a novel collagen resource and make fuller use of the fish catch. Studies have suggested that type I collagen molecules can self-assemble into novel structures when treated at the correct temperature, pH level, and ionic strength [8]. The novel structures that emerge may find applications in the preparation of biomaterials and other functional materials [9]. Fessler et al. confirmed that the amino acid composition of collagen alpha chains varied according to the fish species and the tissues selected, forming different microstructures in the self-assembly process [10].

In this study, ultraviolet (UV) analysis, sodium dodecyl sulfate-polyacrylamide gel electrophoresis (SDS-PAGE), Fourier transform infrared spectroscopy (FTIR), and differential scanning calorimetry (DSC) were used to characterize the collagens extracted from grass carp and crucian carp, which are the most abundant freshwater fish species in China. The assembly properties and morphology of collagens from different sources were investigated during self-assembly curves and scanning electron microscopy (SEM).

## 2. Materials and Methods

### 2.1. Materials

The skins of crucian carp and skins and scales of grass carp were purchased from a local market in Ya-an, China. These were chilled and transported to the laboratory within 30 min of purchase, then immediately washed with cold distilled water. The cleaned skins were cut into 0.5 × 0.5 cm samples using a scalpel. The skin samples and cleaned scales were stored at −20 °C until collagen extraction was carried out. All reagents used were of analytical grade.

### 2.2. Extraction of Collagens

Fat was removed from the skins using 10% n-butyl alcohol (at a solid to solution ratio of 1:20 *w/v*) for 24 h. The solution was changed every 8 h. The scales were decalcified with 0.1 M Na_2_CO_3_ (at a solid to solution ratio of 1:10 *w/v*) for 6 h then placed in a 6% citric acid solution (at a solid to solution ratio of 1:15 *w/v*) for 4 h. For the removal of non-collagenous proteins and pigments, the defatted skins and decalcified scales were mixed with 0.1 M NaOH (3% NaCl and 1% H_2_O_2_) at a ratio of 1:20 (*w/v*) for 6 h. The mixture was then filtered using double layer gauze and washed with distilled water. After drying in air, the processed skins and scales were swelled with 0.5 M acetic acid.

Collagens were extracted by adding 2000 U/g of porcine pepsin (BR, 1:10,000) to an acetic acid solution at 4 °C. The minced skins and scales were stirred continuously in the solution and then filtered thoroughly. The filtrate was centrifuged at 10,000 r/min for 30 min, at the same temperature of 4 °C, and then NaCl was added to bring the final concentration of the suspensions to 2.5 M. These were stored for 24 h to allow the collagens to precipitate. The precipitates were centrifuged at 10,000 r/min for 20 min at 4 °C, then re-dissolved in 0.5 M acetic acid. The extracted collagens were dialyzed against 0.1 M acetic acid for 1 day and deionized water for 2 days with the solution replaced every 8 h, and they were then lyophilized [11,12]. 

### 2.3. UV Absorption Spectra

Collagen samples were prepared by dissolving the lyophilized collagen in 0.5 M of acetic acid at 1 g/L. The ultraviolet absorption spectra of the collagen were recorded using a spectrophotometer (Model UV-1800 PC, Shanghai MAPADA Instrument Co., Ltd., Shanghai, China), in the 190–400 nm range.

### 2.4. FTIR

Freeze-dried 1 mg collagen samples were mixed with 100 mg of KBr, then pressed into disks using a powder-compressing machine (YP-2, Shanghai Shanyue Science Instrument Co., Ltd., Shanghai, China). These were used for IR spectrum recording. The infrared spectra of the collagens were recorded using an FTIR spectrophotometer (Model NICOLET IS10, Thermo Fisher Scientific, Madison, WI, USA). Spectra in the range of 650–4000 cm^−1^ were collected from 32 scans with automatic signal gain at a resolution of 4 cm^−1^. Tests were conducted at a room temperature of 25 °C.

### 2.5. SDS-PAGE Analysis

Type classification and purity determination was conducted using SDS-PAGE with a 12% separating gel (*w/v*) and a 5% stacking gel (*w/v*). The collagens were dissolved in 10 mL of ultrapure water to a concentration of 1 g/L and then mixed with a sample loading buffer (0.5 M Tris-HCl of pH 8.0, containing 20% glycerol, 4% SDS, and 0.1% bromophenol blue) at a ratio of 1:1 (*v*/*v*) in the presence of 10% 2-β-mercaptoethanol. The mixtures were kept in boiling water for 10 min before being centrifuged, and 10 μL of supernatant were loaded into each lane. Electrophoresis was performed at 80 V for the stacking gel and 120 V for the separating gel. After electrophoresis, the gel was stained for 2 h using a 0.1% Coomassie Brilliant Blue R250 solution (a protein stain more sensitive than amido black) and washed with a mixture of 15% ethanol, 7.5% acetic acid, and 77.5% distilled water until the bands became clear. Gels were imaged using an image scanner (Gel Doc XR+, Bio-RAD, Hercules, CA, USA).

### 2.6. DSC

A DSC analysis was performed on a differential scanning calorimeter at the indium standard (Q200, TA Instrument Co., Ltd., New Castle, DE, USA). The lyophilized collagen (2–3 mg) was placed in aluminum pans, sealed, and weighed accurately, with an empty pan used as the reference. Samples were scanned at 2 °C/min over the temperature range of 20–60 °C and under a nitrogen flow of 20 mL/min. The maximum transition temperature (*T*_max_) was estimated from the DSC thermogram.

### 2.7. Self-Assembly of Collagen in Vitro

All procedures were performed following [13], with slight modifications. Lyophilized collagen (100 mg) was dissolved in 0.5 M of an acetic acid solution (100 mL) and stirred for 48 h at 4 °C. The collagen solution was then dialyzed against a phosphate buffer (pH 7.0) containing 150 mM of NaCl. After centrifugation at 2000 r/min for 10 min, the supernatant was incubated in a water bath at 30 °C. The self-assembly kinetics of the different collagens were monitored from the absorbance at 310 nm, using a UV spectrophotometer (Model UV-1800PC, Shanghai MAPADA Instrument Co., Ltd., Shanghai, China).

### 2.8. Measurement of Collagen Fibril Formation

A collagen solution was prepared as described above. After incubation at 30 °C for 10 h, the solutions were centrifuged at 20,000 r/min for 30 min. The hydroxyproline content of the supernatant was measured following the method of Friess et al. [14]. The rate of collagen fibril formation was derived from the percentage decrease in collagen concentration of the supernatant.

### 2.9. SEM

After self-assembly for 10 h at 30 °C, the solution was deposited onto a mica sheet. Samples were dehydrated in an ethanol series in stepwise concentrations of 30%, 50%, 70%, 90%, and 100%, and then they were dried by critical point drying using carbon dioxide. The dried samples were affixed to copper stubs and gold coated. The collagen was detected using SEM (JSM-7500F, JEOL Ltd., Tokyo, Japan).

## 3. Results and Discussion

### 3.1. UV Absorption Spectra of the Extracted Collagen

The UV absorbance of proteins is primarily determined by the molecular structure, including the peptide bonds and side chains. Triple-helical collagen has a maximum peak at 230 nm [15], due to the presence of glycine, proline, and hydroxyproline. Figure 1 shows that the skin of grass carp (GCC), the scales of grass carp (GSC), and the skin of crucian carp (CCC) exhibited maximum absorbance peaks at 235, 235, and 234 nm, respectively. These were close to the peaks of collagens extracted from the scales of red drum fish [16] and from the skin of catla and rohu [17]. In general, the maximum absorption peak of protein is at 280 nm. The absence of absorbance or weak absorbance at 250–280 nm suggested that all three collagens lacked aromatic amino acids, such as tyrosine and phenylalanine, which are sensitive chromophores that absorb UV light at 283 nm and 251 nm, respectively [18]. The absorbance observed at 200–220 nm was attributed to structural materials such as –COOR or –COOH.

### 3.2. FTIR Spectroscopy 

FTIR spectroscopy has been used to study changes in the secondary structure of collagen [19]. The FTIR spectra (650–4000 cm^−1^) of the three collagen types are given in Figure 2. Five typical type I collagen bands were observed: Amide A, amide B, amide I, amide II, and amide III.

The amide A band is contributed mainly by the N–H stretching vibration, and its absorption peak appears in the range of 3400–3440 cm^−1^ [20]. When the NH group of a peptide is involved in a hydrogen bond, the peak is shifted to a lower frequency [21]. The observed amide A bands of the GCC, GSC, and CCC collagen were below this range, at 3322, 3323, and 3331 cm^−1^, respectively. Amide B bands were observed at 2925 cm^−1^ (GCC), 2927 cm^−1^ (GSC), and 2930 cm^−1^ (CCC), corresponding to the asymmetrical stretching of CH_2_ [22]. The amide B of GCC was also detected at 2854 cm^−1^, representing the symmetrical stretching of CH_2_ [22]. 

The amide I band is formed by the C=O stretching vibration of the protein polypeptide backbone, and its characteristic absorption frequency is between 1600 and 1700 cm^−1^. Amide I bands of GCC, GSC, and CCC were found at wavenumbers of 1659, 1660, and 1651 cm^−1^, respectively. This is a sensitive region for protein secondary structure change, and it has the strongest absorption [23]. The amide II band represents N–H bending vibrations coupled with C–N stretching vibrations [24]. The normal absorption range of amide II is between 1500 and 1600 cm^−1^. Amide II bands of GCC and CCC were detected at wavenumbers of 1547 and 1537 cm^−1^, rather than the 1557 cm^−1^ of GSC. This may be due to the hydrogen bonds of GCC and CCC being stronger or more numerous [18]. The amide III peak (1200–1360 cm^−1^) is complex, with the intermolecular interactions of collagen comprising components from C–N stretching and N–Hin-plane bending from the amide linkages, as well as absorption arising from the wagging vibrations of CH_2_ [25]. The amide III bands of GCC, GSC, and CCC were observed at 1238, 1237, and 1237 cm^−1^, respectively. The absorption ratios between amide III and 1454 cm^−1^ were 1.02, 1.01, and 1.04 (all within the range of 1–1.1), which suggested that the triple-helical structure of the collagen was intact [18]. Amide III can be used to determine the triple helix structure of collagen by comparing the absorption peak of amide III with that of 1454 cm^−1^. In general, the ratio of 1 indicated the triple helical structure. The same method has been used frequently in the many papers, such as Ahmad et al. [24] and Matmaroh et al. [25]. The conclusion is similar to that of Kittiphattanabawon et al., who indicated that the triple helix structure of collagen is complete [26].

### 3.3. SDS-PAGE

Collagens are composed of at least two different alpha chains (alpha 1 and alpha 2) and the dimer beta chain (formed by intramolecular cross-linking) that is typical of type I collagen [27]. The SDS-PAGE patterns of GCC, GSC, and CCC are shown in Figure 3. It was found that these three collagens, with the approximately molecular mass of 120 KDa, contained two different α chains (α1 and α2), and the band intensities of the α1-chains were approximately twice those of the α2-chains. This is similar to the conclusion of Muyonga et al. [28]. It seems that collagen exists as a trimer consisting of two α1 and one α2 chains [29]. This is a typical type I collagen, which is the main collagen in dermal tissue [30]. This suggested that collagen consists of at least two different α chains (α1 and α1 chains) and their crosslinked intramolecular dimers (β chains), which are the typical characteristics of type I collagen [16]. The β-components and γ-components had higher molecular weights of 200 KDa. However, no other electrophoretic bands were detected under the alpha 2 chains. This suggested that the extracted collagens maintained their basic structure without containing hydrolyzed small molecule proteins or peptides. Figure 3 shows the SDS-PAGE patterns of GCC, CCC, and GSC. It can be seen that all had bands of α1, α2, and β chains, suggesting that the extracts were typical type I collagens [29]. 

### 3.4. DSC Thermograms

In the thermal conversion of collagen, the collagen triple helix disintegrates into a random coil through a series of physical changes in viscosity, sedimentation, diffusion, light scattering, and other characteristics [31]. The DSC thermograms of the extracted collagens are shown in Figure 4. The maximum temperature reached by GSC was 34.99 °C, lower than that of skin collagens, consistent with the findings of a previous study [32]. However, the maximum temperatures of GCC and CCC were 39.75 and 39.05 °C, respectively. These were slightly higher than the maximum temperatures of 34.99 °C reported for collagens from catla skin and 35.19 °C for collagens from rohu skin by Pal et al. [15], and they were similar to the 39.6 °C reported for catfish skin by Singh, Benjakul, Maqsood and Kishimura [33]. There are two possible reasons. Firstly, the intramolecular hydrogen bonds that stabilize the triple helix structure of collagen may break into several levels in the presence of acetic acid, contributing to the repulsion of collagen molecules in an acidic solution [34]. The collagens from catla skin and rohu skin investigated by Pal were dissolved in acetic acid, whereas in Singh’s research on catfish skin collagen, GCC and CCC were dissolved in deionized water. Second, the thermal stability of collagen depends mainly on its amino acid content, the body temperature of the fish species, and the temperature of the habitat [35,36].

There are many studies on the extraction of collagen from waste materials. Zhang et al. studied the extraction process of collagen from the skin of four freshwater fishes, including tilapia, bighead carp, grass carp and crucian carp, using an acid method. The results showed that the best content of collagen was bighead carp skin, while the content of collagen in grass carp skin was only 26.08%. However, ether degreasing and temperature have great effects on the extraction and yield of collagen after acetic acid treatment [34]. Liu et al. used acid and enzymatic methods to extract acid-soluble and enzymatic-soluble collagen type I from fish scales. SDS-PAGE electrophoresis showed that the electrophoretic band of collagen was the same as the standard type I, and the extracted product was the typical collagen. An amino acid analysis showed that the thermal stability of collagen extracted by the acid method (32.3 °C) was better than that of collagen extracted by enzyme (27.8 °C). If pepsin is added, the solubility of collagen can be increased [35]. These show that the extraction method and pretreatment have great influence on the extraction of collagen. 

Some studies have also suggested that the thermal stability of marine collagen is generally lower than the thermal stability of mammalian collagen. In addition, the thermal stability of collagen is also directly related to the environment and body temperature of the organism [36,37]. The same conclusion was also reported by Nagai et al. Their study showed that the denaturation temperature of collagen from squid skin is 27 °C, which is about 10 °C lower than that of porcine collagen [38]. The denaturation temperatures of the GCC, GSC and CCC collagen extracted in this experiment were 39.75, 34.49 and 39.05 °C, respectively, which were higher than the denaturation temperature of the carp skin. This is not only related to the extraction method and pretreatment, but it is also related to the content of sub-amino acid. It has been speculated that squid skin contains lower sub-amino acid content [39].

### 3.5. Self-Assembly of Collagen

The self-assembly curves of type I collagens from grass carp skin, grass carp scales, and crucian carp skin are shown in Figure 5a. Fibril formation was monitored from the increase in turbidity at 310 nm. The self-assembly curves have been reported to comprise three phases. The first is a lag phase in which the turbidity does not change and nucleation of the collagen fibrils takes place. The second is a growth phase, characterized by a rapid increase in turbidity, in which fibrils are self-assembled by the collagen. The final phase is one of maturity, characterized by stable turbidity and reflecting the formation of three-dimensional networks of fibrils [6]. As can be seen from Figure 5a, GCC and CCC had nucleation periods of 10 min. Similar curves were reported for collagens from silver-line grunt [39] and bester sturgeon [9]. However, GSC had a short lag phase followed by a rapid increase in turbidity, suggesting that this collagen type nucleated more rapidly than tilapia collagen [40] or bovine dermal collagen [41]. The results suggested that GCC, GSC, and CCC were able to assemble spontaneously, which further confirmed that all three maintained their molecular integrity and did not become denatured [42]. Because GSC was converted to gel during the self-assembly process, the change in turbidity of GSC was greater than that of GCC and CCC. A previous study reported that fibrils became longer, thinner, and more flexible as the self-assembly temperature increased, which should be below the denaturation temperature [6].

The rates of collagen fibril formation were assessed after 10 h of fibrillogenesis and are shown in Figure 5b. Collagens from the grass carp skin and scales had similar degrees of self-assembly (GCC 28%, GSC 27.33%), but both were slightly lower than that of CCC (40%). The change in turbidity of the GSC was significantly higher than those of CCC and GCC (Figure 5a), which may due to differences in the self-assembly microstructures. Rasheeda et al. suggested that the self-aggregation rate of type I collagen extracted from rat tendons using an acid method can reach about 90%, which is different from ours and may have been caused by different extraction methods and collagen sources. It can be seen that the collagen content of mammals is superior to that of oviparous animals, and the proper addition of vanillic acid can increase the self-aggregation rate of collagen. Concentration is also important factor for the self-aggregation ability of collagen [43]. The higher the concentration used, the more accumulation of fibers and the higher the degree of self-aggregation obtained. Yan et al. also used the acid method and the enzymatic method to extract collagen from tilapia skin, and acid-soluble and enzymatic collagen all had self-aggregation abilities, which were 18.07% and 19.59%, respectively [40]. In our study, the self-aggregation rate of the GCC, GSC and CCC collagen extracted by the acid–enzyme binding method was superior to the self-aggregation rate of tilapia skin collagen in some literature. The method inconsistently maintains the collagen structure, and the collagen self-aggregation rate is different; the self-aggregation rate of CCC was the highest (40%). Moreover, the extraction temperature also has some effects on the self-aggregation ability of collagen. The results of Yan et al. showed that the 25-degree extracted collagen had a self-aggregation ability, but it was weak—only 6.39%. Collagen extracted at 35 and 45 °C did not have a self-aggregation ability, and a higher extraction temperature could lead to the degradation of collagen subunits, resulting in a decreased collagen self-aggregation ability or even a loss of the self-assembly ability. The extraction temperature was 4 °C, and the self-aggregation ability of collagen could reach to 40% [40]. In summary, the self-aggregation ability of collagen is closely related to collagen source, extraction method, concentration, temperature and added reagents.

### 3.6. The Morphology of Collagen Films

SEM images of the microstructures of GCC, GSC, and CCC before and after self-assembly are shown in Figure 6. The structural appearance of unordered fibrils in GCC (Figure 6a), GSC (Figure 6c), and CCC (Figure 6e) suggested that fibrillogenesis had taken place in the collagens extracted from all three sources. Before self-assembly, GCC and GSC had a porous appearance (Figure 6b,d), whereas CCC appeared as a dense film (Figure 6f). The self-aggregated collagen images (Figure 6a,c,e) shown more porous compared to the images of without self-aggregated collagen (Figure 6b,d,f), and this difference suggested that the self-assembly process had significantly altered the ultra-structure of the collagen. An intertwined fibril network was present in all collagens extracted within the same assembly environment, and this is shown in Figure 6a,c,e. The self-assembled GSC exhibited a greater number of fibrils than CCC or GCC. A clear characteristic D-periodicity was observed in the fibrils of GSC (Figure 6c, red circle), suggesting that the collagen had preserved its structure. Compare to Figure 6c,d, it could be seen that collagen self-aggregation requires external force to promote formation, and the external force of this experiment is a salt solution. Comparing the GCC, GSC, and CCC collagen self-aggregation images, it was known that periodic ring-shaped D bands were observed only in the image of GSC, indicating that different collagens require different abilities and external forces. Figure 5b shows that both GCC and CCC had collagen self-aggregation, but the degree of aggregation was very weak. This result showed that the GCC and CCC collagen require stronger external forces to form a more pronounced periodic ring D band (Figure 6a,c,e). Different extraction methods and extraction temperatures also affect the structure and self-aggregation properties of collagen [42]. This suggests potential applications in the fields of biology and materials science [43]. These differences in the characteristics of the collagen fibrils may partly reflect their different sources [17]. Our results suggested that the in vitro self-assembly of fibrils by GCC, GSC, and CCC is possible if the collagen solution is adjusted appropriately. Self-assembled collagen fibrils can provide docking sites for proteoglycans. Moreover, many in vitro studies on collagen have shown that self-aggregation can provide strong mechanical and thermal stability for collagen matrixes [9]. This has indicated that the in vitro application of GSC will be superior to the application of GCC and CCC. The intertwined fibril network can be widely used in biological and pharmaceutical applications [44,45,46].

## 4. Conclusions

Collagens extracted from grass carp skin (GCC) and scales (GSC), and from crucian carp skin (CCC), were identified as being of type I, with a well-maintained triple-helical structure. GCC and CCC had similar denaturation temperatures that were slightly higher than that of GSC. An SEM analysis confirmed that GCC, GSC, and CCC were all able to form fibrils with distinct structures at 30 °C and a near neutral pH. CCC showed the greatest collagen fibril formation, while GSC formed clear fibrils with a characteristic D-periodicity. This fibril structure may be used to enhance the properties of collagen-based biomaterials, and could support the application of GCC, GSC, and CCC in the development of biomaterials, pharmaceutical materials, and food-packaging materials.

## Figures and Tables

**Figure 1 foods-08-00396-f001:**
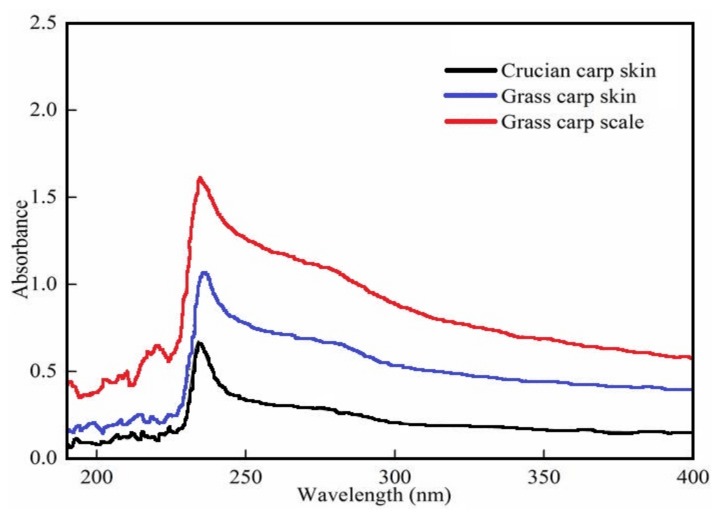
UV spectrum of type I collagen from grass carp skin, crucian carp skin and grass carp scales.

**Figure 2 foods-08-00396-f002:**
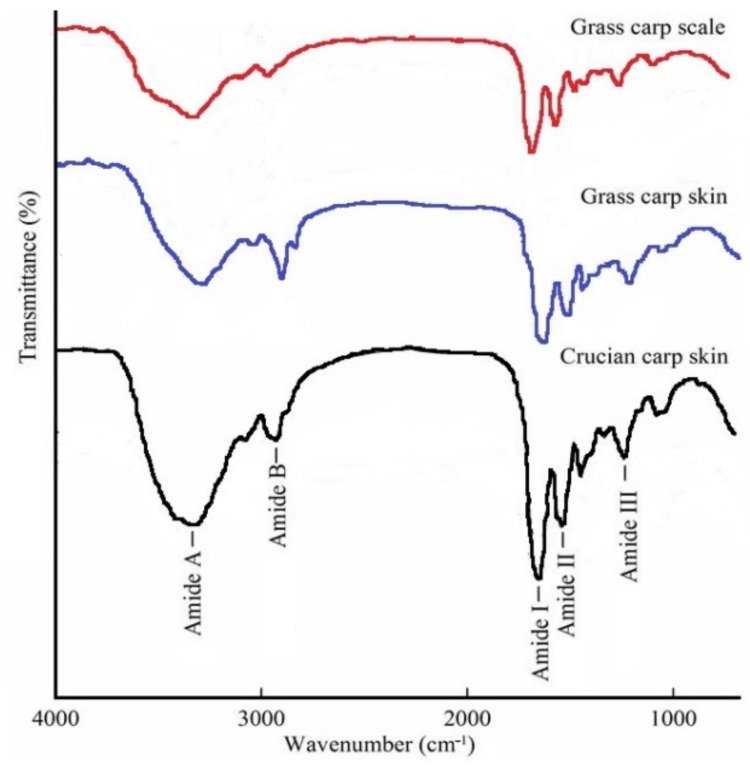
Fourier-transform infrared spectra of type I collagens from grass carp skin, grass crap scales, and crucian carp skin.

**Figure 3 foods-08-00396-f003:**
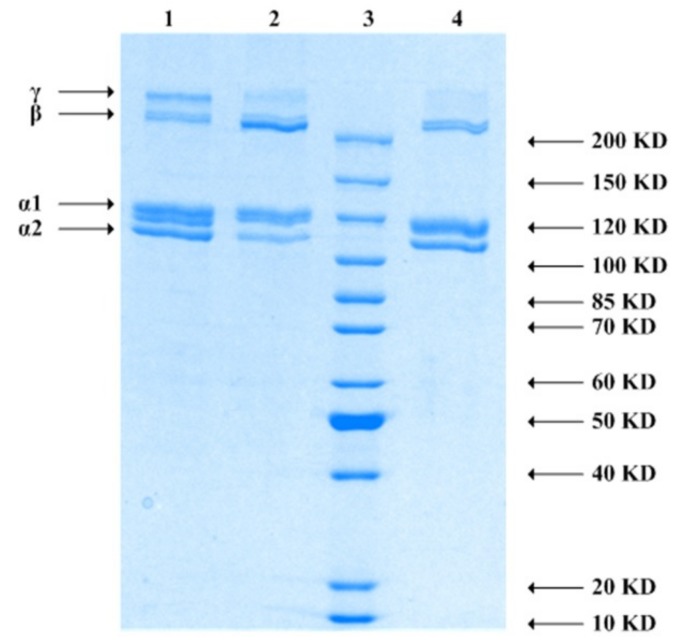
SDS-PAGE patterns of collagens. Lane 1: The skin of grass carp and (GCC), lane 2: The skin of crucian carp (CCC), lane 3: ladder, lane 4: The scales of grass carp (GSC). α: α chain, β: β chain, α1: α1 chain, α2: α2 chain.

**Figure 4 foods-08-00396-f004:**
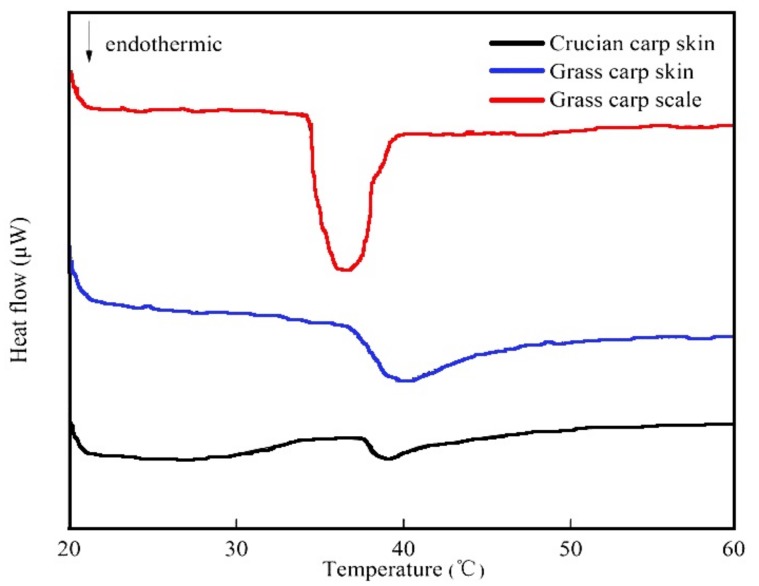
Differential scanning calorimetry (DSC) thermogram of collagens from grass carp skin, scales and crucian carp skin.

**Figure 5 foods-08-00396-f005:**
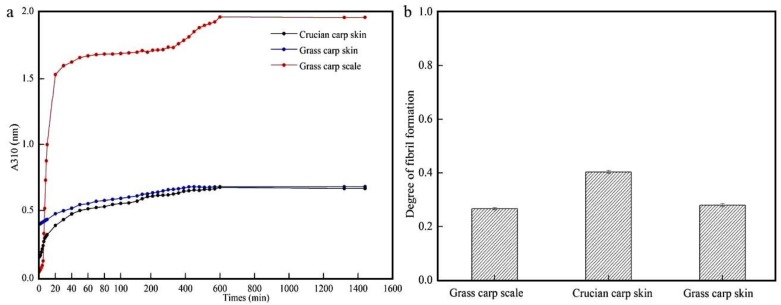
(**a**) Rate of self-assembly and (**b**) degree of fibril-forming of GCC, GSC and CCC.

**Figure 6 foods-08-00396-f006:**
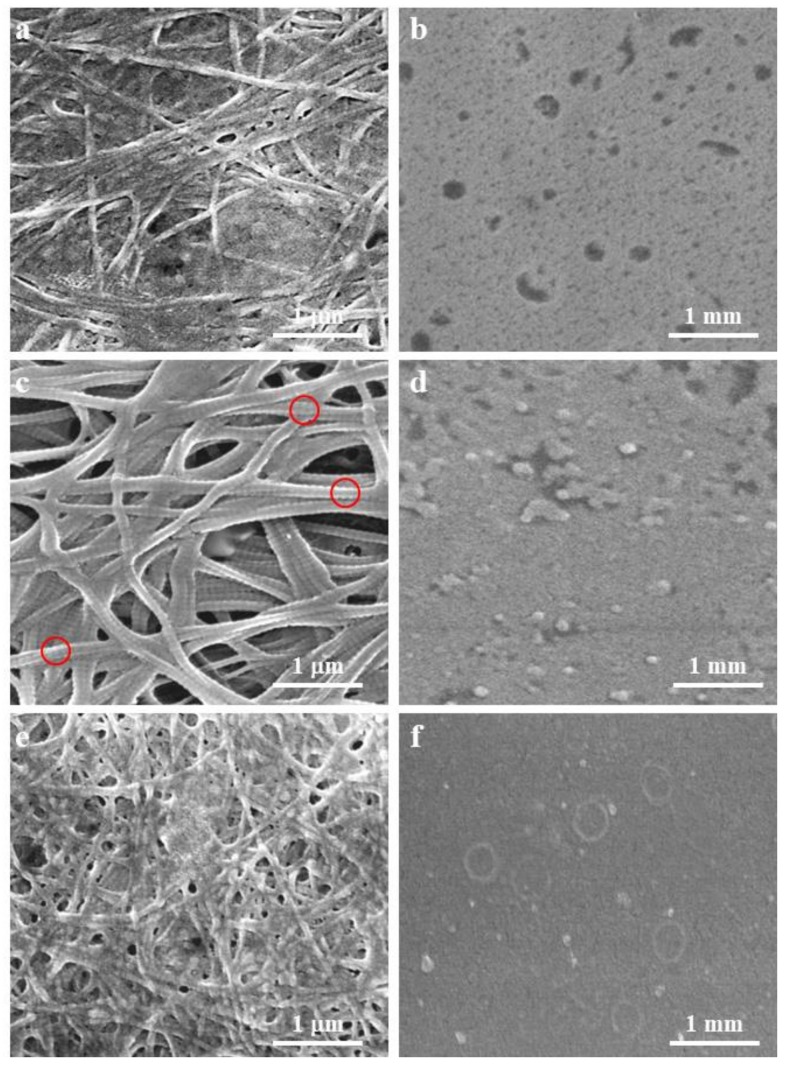
SEM images of collagens from grass carp skin, grass carp scales, and crucian carp skin. (**a**), (**c**) and (**e**) show the SEM images of the GCC, GSC, and CCC collagen after self-assembled for 10 h in a 30 °C salt solution. (**b**), (**d**) and (**f**) show the SEM images of the GCC, GSC, and CCC collagen after drying at room temperature. The red circles indicate the periodic ring D band of collagen.

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
