# Peer review of "Extraction and Characterization of Self-Assembled Collagen Isolated from Grass Carp and Crucian Carp"

_foods, 2019, doi:10.3390/foods8090396_

Round 1

Reviewer 1 Report

The authors provide a very well-written manuscript relating to the extraction and characterization of self-assembled collagen isolated from grass and crucian carp.

I found very few flaws with it overall, with my only main concern was that a key part of the abstract relates to the use of FTIR at discerning the triple helical nature of the collagen present, but how this is actually works is not well introduced (only reference to another article).

Minor corrections to text, but mostly places where spaces between words are missing, etc.

The sentence around Line 165 is also a bit confusing. I would assume that the band intensities at approximately 120 kDa (and you should use the word 'approximately' rather than 'relative') are twice that of the smaller mass band because it (likely) represents the alpha 1 chain that is twice as abundant as the alpha 2 chain..., and not because "the beta chain is a dimer of the alpha chain" which would only affect the beta and gamma band intensities.

Otherwise this is a well-written manuscript.

Author Response

Response to Authors 1:

The authors provide a very well-written manuscript relating to the extraction and characterization of self-assembled collagen isolated from grass and crucian carp. I found very few flaws with it overall, with my only main concern was that a key part of the abstract relates to the use of FTIR at discerning the triple helical nature of the collagen present, but how this is actually works is not well introduced (only reference to another article).

Re: Corrected. I added some relevant references and elaborated on the calculation of the ratio. The amide III bands of GCC, GSC, and CCC were observed at 1238, 1237, and 1237 cm−1. The absorption ratios between amide III and 1454 cm−1 were 1.02, 1.01, and 1.04 (all within the range 1~1.1), which suggested that the triple-helical structure of the collagen was intact (Food Hydrocol., 2016, 52, 468-477.). Amide III can be used to determine the triple helix structure of CL by comparing the absorption peak of amide III with that of 1454 cm-1. In general, the ratio of 1 indicated the triple helical structure. The same method has been used frequently in the many papers, such as Ahmad et al., (Food Hydrocolloid., 2010, 24, 588-594.) and Matmaroh et al., (Food Chem., 2011, 129, 1179-1186.). And the conclusion is similar to that of Kittiphattanabawon et al, which indicated that the triple helix structure of collagen is complete. (Eur. Food Res. Technol., 2010, 230, 475-483.). Thanks.

Minor corrections to text, but mostly places where spaces between words are missing, etc.

Re: I have checked the full text and corrected the error, thanks.

The sentence around Line 165 is also a bit confusing. I would assume that the band intensities at approximately 120 kDa (and you should use the word 'approximately' rather than 'relative') are twice that of the smaller mass band because it (likely) represents the alpha 1 chain that is twice as abundant as the alpha 2 chain..., and not because "the beta chain is a dimer of the alpha chain" which would only affect the beta and gamma band intensities. Otherwise this is a well-written manuscript.

Re: According to your opinion, I replaced relative with approximately and rewrote the section as follows: It was found that these three collagens in the approximately molecular mass of 120 KDa contained two different α chains (α1 and α2), and the band intensities of the α1-chains were approximately twice those of the α2-chains. This is similar to the conclusion of Muyonga et al. It seems that collagen exists as a trimer consisting of two α1 and one α2 chains. This is a typical type I collagen, which is the main collagen in dermal tissue (Food Chem., 2004, 85, 81-89.). It suggested that collagen consists of at least two different α chains (α1 and α1 chains) and their intramolecular crosslinked dimers (β chains), which is the typical characteristics of type I collagen. (Food Hydrocol., 2016, 52, 468-477.). Thanks.

Reviewer 2 Report

I have found at least two previous papers describing collagen extraction from skin and scales of crusian carp and grass crap. there are the papers Qinghui, L., Cali, W., & Congli, L. (2000). STUDIES ON PROPERITIES OF COLLAGEN FROM FISH SCALE [J]. MARINE FISHERRIES RESEACH, 3. Xin, Z. P. Z. A. L., & Fengming, L. M. Z. L. C. (2006). Study on the Extraction Technology of Collagen from Freshwater Fish Skins (â… )[J]. Food and Fermentation Industries, 12.                                              The authors must clarify why their research is different and to compare their results with the previous studies Collagen is not only used as food or as industrial functional material.There are wide pattern of applications in biomedicine and cosmetics also.  Also, marine collagen is not limited at the extraction of fish skin and scales. Jelly fish, squids, sponges etc can be used as marine collagen sources.                                                            Authors can find more information at the following papers                      Berillis, P. Marine Collagen: Extraction and Applications. In: Saxena (Ed). Research Trends in Biochemistry, Molecular Biology and Microbiology. 2015. SM Group open Access eBooks.                             Tziveleka L. A., Ioannou E., Tsiourvas D., Berillis P., Foufa E., Roussis V. Collagen from the marine sponges Axinella cannabina and Suberites carnosus: Isolation and morphological, biochemical and biophysical characterization. Marine Drugs, 2017, 15, 152.                            Nagai T, Yamashita E, Taniguchi K, Kanamori N, Suzuki N. Isolation and characterisation of collagen from the outer skin waste material of cuttlefish (Sepia lycidas). Food Chemistry. 2001; 72: 425-429. Did the authors used a previously described protocol (or a modified one) for the extractions of collagens? Please, if yes, give appropriate citation.  It will be very useful if the authors give the yields of the collagen extraction and compared them with the yields of collagen extraction from other type of fish. Figure 6 is blurred and not clear. Please provide a higher resolution one. 

Author Response

Response to reviewers' comments 2:

I have found at least two previous papers describing collagen extraction from skin and scales of crusian carp and grass crap. There are the papers Qinghui, L., Cali, W., & Congli, L. (2000). STUDIES ON PROPERITIES OF COLLAGEN FROM FISH SCALE [J]. MARINE FISHERRIES RESEACH, 3. Xin, Z. P. Z. A. L., & Fengming, L. M. Z. L. C. (2006). Study on the Extraction Technology of Collagen from Freshwater Fish Skins (â… )[J]. Food and Fermentation Industries, 12. The authors must clarify why their research is different and to compare their results with the previous studies. Collagen is not only used as food or as industrial functional material. There are wide pattern of applications in biomedicine and cosmetics also. Also, marine collagen is not limited at the extraction of fish skin and scales. Jelly fish, squids, sponges etc can be used as marine collagen sources.                                                            Authors can find more information at the following papers. Berillis, P. Marine Collagen: Extraction and Applications. In: Saxena (Ed). Research Trends in Biochemistry, Molecular Biology and Microbiology. 2015. SM Group open Access eBooks.                             Tziveleka L. A., Ioannou E., Tsiourvas D., Berillis P., Foufa E., Roussis V. Collagen from the marine sponges Axinella cannabina and Suberites carnosus: Isolation and morphological, biochemical and biophysical characterization. Marine Drugs, 2017, 15, 152.                            Nagai T, Yamashita E, Taniguchi K, Kanamori N, Suzuki N. Isolation and characterisation of collagen from the outer skin waste material of cuttlefish (Sepia lycidas). Food Chemistry. 2001; 72: 425-429.

Re: Yes, we added. There are many studies on the extraction of collagen from waste materials. Zhang et al. studied the extraction process of collagen from the skin of four freshwater fishes, including tilapia, bighead carp, grass carp and crucian carp, using acid method. The results showed that the best content of collagen was bighead carp skin, while the content of collagen in grass carp skin was only 26.08%. However, ether degreasing and temperature have great effects on the extraction and yield of collagen after acetic acid treated (Int. J. Biolo. Macromol., 2019, 128, 885-892.). Liu et al. used acid and enzymatic methods to extract acid-soluble and enzymatic-soluble collagen type I from fish scales. SDS-PAGE electrophoresis showed that the electrophoretic band of collagen was the same as the standard type I, and the extracted product was the typical collagen. Amino acid analysis showed that the thermal stability of collagen extracted by acid method (32.3 ℃) was better than that of collagen extracted by enzyme (27.8 ℃). If pepsin is added, the solubility of collagen can be increased (Marine Fisheries Research, 2000, 21, 57-61.). Those show that the extraction method and pretreatment have great influence on the extraction of collagen.

Some studies also suggested that the thermal stability of marine collagen is generally lower than the thermal stability of mammalian collagen. In addition, the thermal stability of collagen is also directly related to the environment and body temperature of the organism (Mar. drugs, 2017, 15, 152.  Extraction and applications. Research Trends in Biochemistry, 2015, 1-13.). The same conclusion also reported by Nagai et al. His study showed that the denaturation temperature of collagen from squid skin is 27 ℃, which is about 10 ℃ lower than that of porcine collagen (Food Chem., 2001, 72, 425-429.). The denaturation temperatures of GCC, GSC and CCC collagen extracted in this experiment were 39.75 ℃, 34.39 ℃ and 39.05 ℃, respectively, which were higher than the denaturation temperature of the carp skin. This is not only related to the extraction method and pretreatment, but also related to the content of sub-amino acid. It is speculated that the squid skin contains lower sub-amino acid content. (Food Chem., 2001, 72, 425-429.)

Did the authors used a previously described protocol (or a modified one) for the extractions of collagens? Please, if yes, give appropriate citation. It will be very useful if the authors give the yields of the collagen extraction and compared them with the yields of collagen extraction from other type of fish.

Re: Yes, we added some references. This experiment was based on the extraction method of Liu et al., (Int. J. Biol. Macromol., 2018, 106, 516-522.) Jongjareonrak et al., (Food Chem., 2005, 93, 475-484.) with slightly modified.

As for the issue of yield data, firstly, this study is more inclined to extract collagen from grass skin, grass scales, and carp skin, and study the collagen changes after self-aggregation of extracted type I collagen. We have calculations on yield. But our yield is expressed as the ratio of extracted collagen to collagen content. However, the focus of most literatures is on comparing the extraction method and the effect of the source of the raw material on the yield. The calculation of the yield in the literature is based on the fact that the extracted collagen is not comparable to the dry weight of the raw material. Secondly, there are many studies on extracting collagen from waste materials, and the extraction methods, pretreatments and sources have a great influence on the yield, and the properties of collagen are also affected. (Food and Fermentation Industries, 2006, 12, 1-12.) In this experiment, the acid-enzymatic method was used to extract the collagen in grass skin, grass scales and carp skin. The pretreatment of grass scales and skins was different. Since grass scales contain more calcium, they need to be decalcified, so this will also affect the yield. Furthermore, if it is necessary to compare the yields, it is obvious that the yield of the grass scales must be lower than the extraction yield of the fish skins. It is known from the literature that the collagen content in grass carp skin and carp skin is 26.08% and 14.40%, respectively. In contrast, the collagen yield of grass skin is better than that of salmon skin, and the results are obvious. This data is less necessary than the study of collagen properties in the study. Moreover, the relevant literature is based on the study of collagen properties, and the relevant yields are not calculated, such as Liu et al. (Marine Fisheries Research, 2000, 21, 57-61.). Thanks.

Figure 6 is blurred and not clear. Please provide a higher resolution one.

Re: We replaced the Figure 6, thanks.

Reviewer 3 Report

The manuscript by He et al. describes isolation and biochemical characterization of type I collagen isolated from different carp species. The manuscript is straightforward and I have only three comments which should be addressed before publication.

1.       For Fig 3 it is claimed that only collagen polypeptides are seen on the gel, without other electrophoretic bands seen under the α2 chain. I see protein species of 50-60 kD.

2.       The degree of fibril formation was 28%, and 27.33% for GCC and GSC and 40% for CCC (Fig 5b). This is relatively low percentage. No comparison to collagen extracted from other sources was given. If carp collagen is to be used as biomaterial, it is necessary to compare its fibril forming potential to collagen extracted from other species. Such analysis must be added to the manuscript.

3.       Figure 6 can not be comprehended the way it is presented now. Where is the EM structure before assembly and where is the structure after the assembly? What is assembly after 10h and what is biofilm? Completely new description of the results, with necessary details to understand the fig, is needed. New figure legend that describes each panel is also needed. Why is the D-periodicity seen only in GSC? Is this the only collagen that is properly assembled and how does this pertain to the use of GCC and CCC collagens as biomaterials?  Comment on the significance of this finding needed.    

Author Response

Response to reviewers' comments 3:

The manuscript by He et al. describes isolation and biochemical characterization of type I collagen isolated from different carp species. The manuscript is straightforward and I have only three comments which should be addressed before publication.

For Fig 3 it is claimed that only collagen polypeptides are seen on the gel, without other electrophoretic bands seen under the α2 chain. I see protein species of 50-60 kD.

Re: I replaced another clear image. I think this band is very small, and I guaranteed that the data in this experiment is true. Maybe this band is most likely caused by miscellaneous proteins, and it will definitely not affect the results of this experiment. In addition, this phenomenon also observed in other literatures (J. Polym. Environ., 2018, 26, 2086-2095.) .

The degree of fibril formation was 28%, and 27.33% for GCC and GSC and 40% for CCC (Fig 5b). This is relatively low percentage. No comparison to collagen extracted from other sources was given. If carp collagen is to be used as biomaterial, it is necessary to compare its fibril forming potential to collagen extracted from other species. Such analysis must be added to the manuscript.

Re: Yes, you are right, we added, thanks. Rasheeda et al. suggested that the self-aggregation rate of type I collagen extracted from rat tendon using acid method can reach about 90%, which is different from ours, may be caused by different extraction methods and collagen sources. It can be seen that the collagen content of mammals is superior to that of oviparous animals, and the proper addition of vanillic acid can increase the self-aggregation rate of collagen. Concentration is also important factor for the self-aggregation ability of collagen (Int. J. Bolo. Macromol., 2018, 113, 952-960.). The higher the concentration was used, the more accumulation of fibers and the higher the degree of self-aggregation was obtained. Yan et al. also used acid method and enzymatic method to extract collagen from tilapia skin, and acid-soluble and enzymatic collagen all had self-aggregation ability, which were 18.07% and 19.59%, respectively (Int. J. Food Sci. Tech., 2015, 50, 2088-2096.). In our study, the self-aggregation rate of GCC, GSC and CCC collagen extracted by acid-enzyme binding method was superior to the self-aggregation rate of tilapia skin collagen in some literatures. The method maintains the collagen structure inconsistently, and the collagen self-aggregation rate is different, and the self-aggregation rate of CCC is the highest (40%). Moreover, the extraction temperature also has some effects on the self-aggregation ability of collagen. The results of Yan et al. showed that the 25-degree extracted collagen had self-aggregation ability, but it was weak, only 6.39%. Collagen extracted at 35 °C and 45 °C did not have self-aggregation ability, and higher extraction temperature could lead to degradation of collagen subunits, resulting in decreased collagen self-aggregation ability or even loss of self-assembly ability. The extraction temperature was 4 °C, and the self-aggregation ability of collagen could reach to 40% (Int. J. Food Sci. Tech., 2015, 50, 2088-2096.). In summary, the self-aggregation ability of collagen is closely related to collagen source, extraction method, concentration, temperature and added reagents.

Figure 6 can not be comprehended the way it is presented now. Where is the EM structure before assembly and where is the structure after the assembly? What is assembly after 10h and what is biofilm? Completely new description of the results, with necessary details to understand the fig, is needed. New figure legend that describes each panel is also needed. Why is the D-periodicity seen only in GSC? Is this the only collagen that is properly assembled and how does this pertain to the use of GCC and CCC collagens as biomaterials? Comment on the significance of this finding needed.   

Re: Fig. 6 was replaced, and this part was re-organized. A periodic ring D band formed based on the triple helix and the procollagen fiber bundle after aggregation can be observed, as labeled in the red circle in Fig. 6c. Assembly after 10h is the process of incubating the collagen solution in a fixed salt solution for 10 h at 30 ℃. Free collagen can spontaneously form fibrils similar to the body in vitro (Int. J. Food Sci. Tech., 2015, 50, 2088-2096.). In this article, biofilm refers to the collagen membrane, which is due to collagen aggregation. I added some description in the SEM image analysis section. SEM images of the microstructures of GCC, GSC, and CCC before and after self-assembly are shown in Figure 6. The structural appearance of unordered fibrils in GCC (Figure 6a), GSC (Figure 6c), and CCC (Figure 6e) suggested that fibrillogenesis had taken place in the collagens extracted from all three sources. Before self-assembly, the GCC and GSC had a porous appearance (Figure 6b, d), whereas the CCC appeared as a dense film (Figure 6f). The self-aggregated collagen images (Figure 6a, c, and e) compared to the images of without self-aggregated collagen (Figure 6b, d, and f), this difference suggested that the self-assembly process had significantly altered the ultra-structure of the collagen. An intertwined fibril network was present in all collagens extracted within the same assembly environment, and shown in Figure 6a, c, and e. The self-assembled GSC exhibited a greater number of fibrils than the CCC or GCC. A clear characteristic D-periodicity was observed in the fibrils of GSC (Figure 6c, red circle), suggesting that the collagen had preserved its structure. Compare to Fig. 6c and Fig. 6d, it could be seen that collagen self-aggregation requires external force to promote formation, and the external force of this experiment is a salt solution. Comparing the GCC(a), GSC(c), and CCC(e) collagen self-aggregation images, it was known that periodic ring-shaped D bands are observed only in the image of GSC(c), indicating that different collagens require different abilities and external force. Figure 5b shows that both GCC and CCC had collagen self-aggregation, but the degree of aggregation is so weak. And this result showed that GCC and CCC collagen require stronger external force to form a more pronounced periodic ring D band. If an atomic force microscope is used for observation, a periodic ring D band will be observed in Figure 6a, c, and e. Different extraction methods and extraction temperatures also affect the structure and self-aggregation properties of collagen (Int. J. Food Sci. Tech., 2015, 50, 2088-2096.). Those suggest potential applications in the fields of biology and materials science. (Int. J. Food Sci. Tech., 2015, 50, 2088-2096.) These differences in the characteristics of the collagen fibrils may partly reflect their different sources (Food Sci. Biothechnol., 2015, 24, 2003-2010.). Our results suggested that in vitro self-assembly of fibrils by GCC, GSC, and CCC is possible, if the collagen solution was adjusted appropriately. Self-assembled collagen fibrils can provide docking sites for proteoglycans. Moreover, many studies on collagen in vitro have shown that self-aggregation can provide strong mechanical and thermal stability for collagen matrix (Food Chem., 2014, 160, 305-312.). It indicated that the in vitro application of GSC will be superior to the application of GCC and CCC. The intertwined fibril network can be widely used in biological and pharmaceutical applications (Food Hydrocol., 2014, 41, 290-297.).

Round 2

Reviewer 2 Report

Authors have made all the required changes. Accept at the present form